# Supercoil Levels in *E. coli* and *Salmonella* Chromosomes Are Regulated by the C-Terminal 35–38 Amino Acids of GyrA

**DOI:** 10.3390/microorganisms7030081

**Published:** 2019-03-15

**Authors:** Nikolay S. Rovinskiy, Andrews A. Agbleke, Olga N. Chesnokova, N. Patrick Higgins

**Affiliations:** Department of Biochemistry and Molecular Genetics, University of Alabama at Birmingham, Birmingham, AL 35294-0024, USA; nikolay.rovinskiy@aol.com (N.S.R.); agbleke@fas.harvard.edu (A.A.A.); ochesnokov@fau.edu (O.N.C.)

**Keywords:** DNA topology is investigated that involves two types of DNA coiling, the first type is right-handed coils that Watson/Crick DNA strands adopt by winding around each other every 10.6 base pairs, the second type involves coiling of the double strands around each other in either a left handed (−) or right handed (+) direction, DNA gyrase is an enzyme with two protein subunits, GyrA and GyrB that catalyzes (−) supercoiling at the expense of ATP hydrolysis, gamma delta (γδ) resolvase is a site-specific recombinase from the γδ transposon the that utilizes (−) supercoils to delete a DNA sequence that is flanked by 100 bp Res sites, gyrase processivity refers to the number of reaction cycles one enzyme carries out in a single DNA binding event, the Q10 rule states that reaction rates double for every 10° C increase in temperature, the GyrA C-terminal domain (CTD) includes a long DNA binding section that has 6 pinwheel elements plus a short 35–38 amino acid terminus called the tail, RNA polymerase (RNAP), the *E. coli* and *Salmonella* condensin is a multi-protein complex composed of three proteins, MukB, MukE, and MukF that compacts chromosomal DNA

## Abstract

Prokaryotes have an essential gene—gyrase—that catalyzes negative supercoiling of plasmid and chromosomal DNA. Negative supercoils influence DNA replication, transcription, homologous recombination, site-specific recombination, genetic transposition and sister chromosome segregation. Although *E. coli* and *Salmonella* Typhimurium are close relatives with a conserved set of essential genes, *E. coli* DNA has a supercoil density 15% higher than *Salmonella*, and *E. coli* cannot grow at the supercoil density maintained by wild type (WT) *Salmonella*. *E. coli* is addicted to high supercoiling levels for efficient chromosomal folding. In vitro experiments were performed with four gyrase isoforms of the tetrameric enzyme (GyrA_2_:GyrB_2_). *E. coli* gyrase was more processive and faster than the *Salmonella* enzyme, but *Salmonella* strains with chromosomal swaps of *E. coli* GyrA lost 40% of the chromosomal supercoil density. Reciprocal experiments in *E. coli* showed chromosomal dysfunction for strains harboring *Salmonella* GyrA. One GyrA segment responsible for dis-regulation was uncovered by constructing and testing GyrA chimeras in vivo. The six pinwheel elements and the C-terminal 35–38 acidic residues of GyrA controlled WT chromosome-wide supercoiling density in both species. A model of enzyme processivity modulated by competition between DNA and the GyrA acidic tail for access to β-pinwheel elements is presented.

## 1. Introduction

Species differences pose serious problems for research in drug design, in developing therapeutic treatments for many human diseases, and for advancing gene therapy protocols that are safe for humans [1]. The basic differences between a man and mouse often involve yet-to-be defined biochemical pathways and responses in the species-specific innate immunological repertoire. With institutional funding’s emphasis on translational medicine and systems biology of all the “omics” (proteomics, transcriptomics, metabalomics and phenomics), little attention is being directed at understanding “core biochemistry” in different species, different cell types and different genetic backgrounds. Species differences in “core biochemistry” can be studied in “simple” bacteria [2]. In 2007, Champion discovered multiple unexpected differences between *E. coli* and *Salmonella* [3]. Four significant mysteries include: 1) Different phenotypes for the identical mutation in GyrB; 2) Different phenotypes for mutants in the MukB subunit of the condensin complex; 3) Toxicity of *Salmonella* GyrB when it is expressed at low levels in *E. coli*; 4) Significant differences in the supercoil density of plasmid and chromosomal DNA in cells growing on Luria Broth (LB) at all temperatures showed that *E. coli* is a “high supercoil organism.” The aim of our work is to explain how supercoil differences are established throughout the chromosome in *E. coli* and *Salmonella* and to define the enzymatic mechanism(s) that coordinates pathway flow of transcription, translation, and protein folding for cells of each organism during exponential growth. 

DNA supercoiling has long been considered to be a global regulatory factor in bacterial gene expression [4,5,6]. Three bacterial topoisomerases play critical roles in equilibrating supercoiling and DNA entanglements caused by replication and RNA transcription [7,8,9]. Topo I, a type IA topoisomerase, removes negative supercoiling in a cofactor-independent reaction that protects DNA from forming toxic R-loops caused by RNA invasion of DNA at hyper-supercoiled regions [10,11]. Gyrase replenishes (–) supercoils depleted downstream of operons [9,12] and Topo IV untangles and decatenates DNA strands and removes (+) supercoils generated during DNA replication and segregation [13,14,15].

Mutations that reduce the supercoil density (σ) of *E. coli* DNA by only 15% are toxic and alter the efficiency and/or fidelity of DNA replication [16], chromosome segregation [17,18], RNA transcription [19,20], homologous and site-specific recombination [16] and gene transposition reactions [21,22,23]. In sharp contrast, wild type (WT) *Salmonella* Typhimurium grows naturally at an average supercoil density 15% lower than *E. coli* [3] and *Salmonella* grows and produces well-formed colonies in strains having a complete loss of diffusible chromosome supercoiling [9,12]. 

We dissected the mechanism that produces these confusing results by analyzing the four *E. coli* and *Salmonella* isoforms of the gyrase tetramer. Each isoform was tested in vitro by measuring supercoil rates and endpoints on a high affinity plasmid substrate. The four isoforms were then tested in vivo in *Salmonella,* and chromosome supercoil densities at 10 positions around the genome were evaluated. The in vivo results proved GyrA to be a prime player. By making *E. coli–Salmonella* chimeras, we identified a critical gyrase control point. The carboxy terminus of GyrA acts like a rheostat to regulate the unique species-specific supercoil needs of highly transcribed and translated regions in chromosomes of both species.

## 2. Materials and Methods

### 2.1. Strains and Cloning

All strains used here were made in our lab for this work and are listed in Appendix A. The WT *gyrA* and *gyrB* genes from *E. coli* (NH3612) and *Salmonella* (NH3358) were cloned into pLIK-HK plasmid (New England Biolabs, Ipswitch NY USA) using ligation-independent cloning protocol (LIC protocol) described by Stols et al [24]. Each cloned protein was fused to an N-terminal 6-histidine tag that was removed by recombinant TEV protease (Sigma Aldrich, St Louis MO, USA) digestion during purification. 

### 2.2. Growth Rate Measurements and Resolution Assays

The cell doubling times of WT and mutant strains were measured in early-mid log phase between the optical cell densities at 600 nm of 0.01 and 0.4. Each strain was tested in triplicate, starting from three independent colonies grown overnight in fresh LB containing 5 g NaCl, 5 g Yeast Extract, and 10 g Tryptone dissolved in 1 L of water. Cells from fresh overnight cultures were diluted 100-fold in fresh LB, and doubling times are reported ± 1 standard deviation from the mean.

To measure resolution efficiencies, log-phase cultures grown at 30°C in LB were tested at a cell density of 50 Klett units. An aliquot (0.1 mL) of each culture was incubated at 42 °C in a shaking water bath for 10 min to induce resolvase expression and then was diluted with 2 mL of LB + Cm for overnight incubation at 32 °C. The next day, 100 µL aliquots of 10^−6^ dilutions of each sample and an un-induced control were plated on NCE glucose minimal medium containing chloramphenicol plus 5-bromo-4-chloro-3-indolyl-d-galactosidase (X-gal) plus 100 µM IPTG [8]. Each data point represents the average ± 1 standard deviation of three independent cultures from which at least 200 colonies were analyzed for LacZ deletions.

### 2.3. Enzyme Purification

Gyrase subunits were cloned using pLIK-HK plasmids (New England Biolabs, Ipswitch, NY, USA) [25] and were expressed in *E. coli* BL21 pLysS [26]. Cells grown in 60 L batches at 30 °C to a cell optical density of 0.5 at 650 nm were induced by adding isopropyl β-D-1-thiolgalactoside (IPTG) to a final concentration of 1 mM Cells harvested after three hours were rapidly chilled to 4 °C and concentrated by centrifugation. Each growth harvest was resuspended in ten 60 mL batches in buffer containing 50 mM Tris-HCl, pH 8.0, 10% glycerol, 1 mM EDTA, 1 mM dithiothreitol (DTT) and frozen at -50° for future use. Lysis in a French pressure cell was followed by removal of cell debris by centrifugation. Streptomycin sulfate (40% solution) was added to a final concentration of 4%, and mixtures were stirred at 4 °C for 30 min. After centrifugation for 30 min in a Beckman J21 rotor at 10,000 rpm, solid ammonium sulfate was added to the supernatant at a final concentration of 70% saturation. The ammonium sulfate pellet was suspended and dialyzed in Ni-NTA column loading buffer. Chromatography on Qiagen Ni-NTA resin (Qiagen, Redwood City, CA, USA) was carried out using buffers provided by the resin supplier. Proteins were loaded onto a Ni-NTA column in phosphate buffer (pH 8.0) containing 10 mM imidazole; the column was washed in buffer containing 20 mM Imidazole; and gyrase subunits were removed from the column by a step elution in phosphate buffer (pH 8.0) containing high imidazole (250 mM). The 6-His affinity tag was removed by TEV protease digestion, and each gyrase subunit was applied to a second Ni-NTA column, where it passed through the resin in 10 mM imidazole buffer. This step eliminated proteins without a HIS tag that bind to the resin. Gyrase subunits were further purified by gel filtration through a Sephacryl S-200 column (Sigma Aldrich, St. Louis MO, USA) in 0.2 M KPO_4_, pH 7.4, 10% glycerol. Fractions with the highest specific supercoiling activity were dialyzed into storage buffer (50 mM Tris-HCl pH 8.0, 100 mM KCl, 1 mM DTT, and 50% glycerol) and stored at −20 °C. Enzyme reconstitution was done by mixing 76 pmol GyrA with 38 pmol of GyrB in 100 µL of storage buffer.

### 2.4. Biochemical Methods

Preliminary experiments were always carried out in which serial dilutions of the each GyrA-GyrB pair to be tested was serially diluted, pre-incubated with relaxed DNA for 30 min at 30 °C, and then incubated after ATP addition for 90 s. The highest dilution that supercoiled all the relaxed substrate was then used for time course studies on the same day. The reason that GyrB is the limiting subunit is that it is present in cells at half the concentration of GyrA [27]. Temperature sensitive (TS) GyrB preparations are especially troublesome; they are easily denatured by transfers out of and back into a −20 °C freezer and must often be purified fresh for critical experiments.

Supercoiling reactions were carried out in G-buffer containing 35 mM Tris-HCl pH 7.4, 18 mM potassium phosphate pH 7.4, 1 mM DTT, 10 mM MgCl_2_, 50 µg/mL bovine serum albumin, 5 mM spermidine-HCl, and 0.4 μg/mL yeast tRNA [27]. Pre-tested gyrase ensembles and relaxed pMP1000 plasmid DNA were mixed in G-buffer and incubated for 30 min at 30 ° C to form stable gyrase-DNA complexes [28,29]. Supercoiling was initiated by mixing in ATP at a final concentration of 1.2 mM, and reactions were quenched with EDTA (10 mM) pH 8.0 and 0.1% SDS. Single dimension agarose gels with 0.48 µM chloroquine were run at 47 V for 20 h, and two-D gels were run for 15 hours at 47 V in the presence of 0.48 µM chloroquine in the first dimension and 20 µM chloroquine in the second. Gel bands were transferred to Zeta-Probe^®^ GT Genomic Tested Blotting Membranes from Bio-Rad^®^ (Hercules, CA, USA). Southern blot hybridization was done using Roche^®^ DIG High Prime DNA Labeling and Detection Starter Kit II (Sigma Aldrich, St. Louis, MO, USA).

### 2.5. Genetic Methods

The *gyrA* and *gyrB* genes of *E. coli* were introduced into the *S.* Typhimurium chromosome using the λ red “recombineering” system [30]. To select for recombinant *gyrB* and *gyrA* genetic exchange, a module encoding resistance to kanamycin or chloramphenicol was inserted within 1 kb of each *E. coli* and *S. Typhimurium* gyrase gene. The drug cassette and linked GyrA or GyrB genes were amplified for recombination into recipient chromosomes as described previously [3,31]. Supercoil sensors were moved into *Salmonella* by P22 transduction as described previously [32].

## 3. Results

### 3.1. Supercoiling Assays of WT and Interspecies Forms of Gyrase

The amino acid differences between *gyrA* and *gyrB* of *E. coli* and *Salmonella* are illustrated in Figure 1. *Salmonella* is the reference sequence, and horizontal black lines show positions that differ in the *E. coli* homologues. Both GyrB proteins are 804 amino acids long, and *Salmonella* GyrA has 738 residues, which is 3 longer than *E. coli* GyrA at 735. GyrA includes 4 conserved structural domains with critical structure/function roles that extend to all known prokaryotic and eukaryotic type II topoisomerases. The last 35 amino acids of *E. coli* GyrA are essential for supercoiling activity in *E. coli* gyrase, but this region varies widely in different bacterial species and is not considered required for supercoiling by *M. tb* [33,34] or *B. subtilis* gyrase [35].

Supercoiling reactions are most efficient when gyrase binds to rare “strong” DNA sequences that promote processive reactions. The highest affinity gyrase DNA site known is the strong gyrase site (SGS) from the center of phage Mu [36,37]. Gyrase forms a complex with SGS DNA that has a dissociation half-life >40 h, and ensembles formed in the absence of ATP can be purified by Sepharose exclusion chromatography [29]. Addition of ATP to pre-formed complexes leads to maximal plasmid supercoiling within 60 s with a supercoil end point greater than that present in plasmid DNA isolated from WT cells where σ equals (−) 0.069 (Figure 2). Single-molecule rotor bead experiments confirm that a single gyrase bound to the Mu-SGS catalyzes processive bursts of >100 supercoils (50 cycles) per min [38].

Quantitative supercoiling speed and supercoil endpoints were measured in vitro with gyrase-DNA complexes bound to the relaxed plasmid pM1000, which has the Mu *nuB103* SGS cloned into a pUC19 poly-linker. Progression from relaxed to supercoiled conformations after ATP addition was measured by electrophoresis in two agarose gel systems containing chloroquine phosphate. Both systems resolve all 25 supercoil steps from relaxed to the fully supercoiled in vivo endpoint. The chloroquine concentration in the gel in Figure 2 removes half of the 25 (−) supercoils from naturally supercoiled pUC19 DNA and introduces 13 (+) supercoils into relaxed plasmid DNA under gel electrophoresis conditions; the corresponding (+) and (–) supercoil isoforms run near each other in the gel. The initial products of supercoiling move higher in the gel up to an ΔLk value of −12, which is near nicked DNA (steps are indicated by red dashes in lane 2 and 4). DNA with ΔLk values > −13 moves progressively faster toward the bottom of the gel, and hyper-supercoiled species run faster than supercoiled DNA marker isolated directly from WT *E. coli* (Figure 2).

Supercoil characteristics of WT *E. coli* and WT *Salmonella* gyrase (A*_2_*_Ec_-B*_2_*_Ec_ and A*_2_*_ST_-B*_2_*_ST_) were also compared with transgenic isoforms A*_2_*_Ec_-B*_2_*_ST_ and A*_2_*_ST_-B*_2_*_Ec_ (Figure 2). Reactions (100 µL) assembled on ice in G-buffer with 230 fmol of relaxed pMP1000 were incubated at 30 °C for 30 min to form stable gyrase-DNA complexes [28]. ATP was stirred into each reaction at 1.2 mM, and 10 µL aliquots were withdrawn and stopped by addition of SDS at 5 s, 10 s, 20 s, 40 s, and 80 s time points. Samples loaded onto an agarose gel with 0.5× TBE and 0.48 µM chloroquine phosphate were subjected to electrophoresis at 47 V for 20 h. DNA bands representing single topoisomers were visualized from DNA Southern blots. 

Time-course clusters for all forms of gyrase are shown in Figure 2. *E. coli* gyrase all blue (A*_2_*_Ec_-B*_2_*_Ec_) is the first cluster. At 5 s, a band overlaps the position of native supercoiled DNA in lane 1. This indicates a linking number change (Lk) of −24 (red hatch marks beside lanes 2 and 3). Thus, a single gyrase bound to one SGS site supercoils DNA in vitro at five supercoils/sec in dilute solution at 30°C. This agrees with single molecule rotor bead experiments, assuming a Q_10_-related 2-fold rate increase for reactions carried out here at 30°C compared to the 20°C room temperature of rotor bead experiments [38]. At the 10 s time point, a cluster of hyper-supercoiled DNA bands migrate further down the gel than supercoiled species in a native supercoiled plasmid control lane 1 (σ = −0.069).

The second cluster had the weakest enzyme. *E. coli* GyrA and *Salmonella* GyrB starts very slowly with relaxed DNA remaining near the starting position for >10 s. It required 20 s to generate a supercoiled population near the density of native plasmid and hyper-supercoiling required nearly 80 s. The other transgenic combination of *Salmonella* GyrA and *E. coli* GyrB in cluster 3 was a better enzyme. This form had kinetics similar to *Salmonella* gyrase in cluster 4. Most DNA reaches the hyper-supercoil zone by 40 s. Since supercoil free energy is an exponential function of linking number [39,40], *E. coli* gyrase is clearly more powerful than *Salmonella* gyrase and both transgene combinations when assayed using the pMP1000 substrate.

To confirm the kinetic rate of 5 supercoils/s, each enzyme was re-tested in 5 s assays where all products are displayed in a 2D gel that unambiguously resolves each topoisomer from relaxed to the hyper-supercoiled state of −30 (Figure 3). The central relaxed topoisomer is marked with a red dot in panel A, and supercoil steps above the dot by band counting are marked at −15, −20, and −25 in panels B–F. Figure 3C shows the 5 sec assay for *E. coli* A_Ec2_-B_Ec2_. Every complex does not initiate supercoiling immediately after ATP addition, so relaxed DNA bands are present along with a complete distribution of topoisomers to the −30 position. *Salmonella* gyrase A_ST2_-B_ST2_ (Figure 3F) yielded DNA product at −25, but the distribution lags behind the *E. coli* enzyme. The transgenic combination of A_Ec2_-B_St2_ (Figure 3D) is again the worst combination, with most DNA remaining at the fully relaxed positions. The transgenic form A_St2_-B_Ec2_ (Figure 3E) showed a pattern similar to *Salmonella* gyrase in F. Overall, the power relationship was A_Ec2_-B_Ec2_ > A_St2_-B_St2_
-> A_St2_-B_Ec2_ >> A_Ec2_-B_St2_.

### 3.2. In Vivo Chromosome Supercoiling in Transgenic Salmonellae

In vitro results in Figure 2 and Figure 3 show that *E. coli* gyrase is faster and/or more powerful than *Salmonella* gyrase. This result alone could explain the higher sustained in vivo supercoil DNA levels for *E. coli* compared to *Salmonella*. To test this prediction, transgenic *Salmonella* strains were created by exchanging *E. coli* homologues for *Salmonella* genes using phage lambda *red* recombineering. Sequence analysis of PCR-amplified chromosomal DNA confirmed the genetic structure of each strain. All strains grew on LB plates incubated at 30 °C and 42 °C. Because transgenic GyrB_ST_ expression is deleterious for *E. coli* [3], we tested each transgenic *Salmonellae* for a detectable growth phenotype by measuring cell doubling times. Three independent colonies of each test strain were incubated in shaking liquid LB cultures at 30°, and the A_600_ was measured by Klett, which allows rapid sampling with minimal manipulation for each measurement. WT *Salmonella* (NH6303) doubling time was 40 ± 1 min (Table 1). *Salmonella* with either a *gyrB_Ec_* transgene (NH6304) or a *gyrA_Ec_* allele (NH6281) had a small but measurably slower doubling time of 43 ± 1 min. The transgenic replacement with both *gyrA_Ec and_ gyrB_Ec_* had the slowest doubling time of 44 ± 1 min.

Due to a small but measurable impact of each transgene on cell division, we measured supercoil densities at 10 positions in the *Salmonella* chromosome. Strains were constructed with each gyrase isotype to analyze in vivo chromosomal supercoiling. This technique exploits the supercoil-dependent resolvase deletion reaction for a 9 kb *lacZ* operon module placed at different loci (see Appendix A) and detects supercoil changes over a 100-fold range [9,32].

### 3.3. Supercoiling in gyrB_Ec_ Transgenics

The replacement of *Salmonella gyrB* with *E. coli gyrB* resulted in changes from WT (Figure 4). However, rather than uncovering a uniform change at all locations, differences from WT supercoil levels varied by location. The chromosome was a patchwork of independent supercoil domains that did not change uniformly. We calculated a Mutation Impact Factor number (MIF) for each locus (Appendix A) which compares the resolution efficiency of the transgenic strains versus the WT. For example, at Cs57.65 (upstream of the *rrnG* operon) the excision efficiency for WT (NH6257) 69 ± 11% increased to 89 ± 3% (MIF = 130) in the transgenic NH6271. This represents a 30% increase in deletion frequency. Downstream of the *rrnG* transcription terminator at Cs 57.64, resolution occurred in 31 ± 11% of WT NH6258, and the transgenic rate was 50 ± 7% (MIF = 160) in NH6272. At position Cs 85 the supercoil sensor lies immediately upstream of the ATP operon, which encodes 9 integral membrane proteins that generate ATP using the proton motive force across the cytoplasmic membrane for energy. The WT *Salmonella* recombination rate of 66±6% NH6259 rose to 78 ± 1% (MIF = 120) for NH6273. Thus, at these positions a transgenic *E. coli* GyrB increased supercoil density of the *Salmonella* chromosome.

But other locations had supercoiling losses. At Cs9 the WT resolution rate of 49 ± 17% in NH6265 dropped to 29 ± 8% (MIF = 59) NH6279), and near *dif* site at Cs 33 the WT rate of 34 ± 9% for NH6265 fell to 13 ± 3% (MIF = 38) in NH6277. The average resolution efficiency (58 ± 15%) for WT was the same for averaged sum at 10 locations in the *gyrB_Ec_* replacements (57 ± 22%). But the global average did not reflect a mass action supercoiled structure; local supercoil environments behaved differently and high transcription regions GyrB_Ec_ increased supercoil density (black numbers) above the level of WT (red numbers).

### 3.4. Salmonella GyrA + GyrB Double Transgenics

The resulting average for strains carrying the *gyrB_Ec_* transgene was in line with the in vitro power analysis of the four gyrase combinations, if one focuses on the average. We naively anticipated that the average supercoil level would rise in *Salmonella* transgenics having both *E. coli* gyrase subunits, based on in vitro results with the plasmid. But a 10% increase of 4 min in cell doubling time (Table 1) presaged significant supercoiling problems throughout the genome (Figure 4, green numbers). At the ATP operon near Cs85, resolution fell from 66 ± 8% in WT NH6259 to 41 ± 10% (MIF = 28) in the transgenic *NH6295*, representing a 72% decline. At the *rrnG*, the upstream rate fell from the WT value in *NH6257* of 69 ± 11% to 31 ± 7% in *NH6293* (MIF = 45). The downstream rate fell from the WT value in *NH6258* of 31 ± 11% to 6 ± 5% (MIF = 20) in NH6294. The average resolution efficiency at ten positions in WT cells of 58 ± 15% dropped by half to 27 ± 15% when both *E. coli* gyrase subunits were present. Interpolation of the graph in Appendix A indicates that *E. coli* gyrase caused the average diffusible supercoil density to drop by 33%, from WT σ = −0.033 to mutant value of σ = −0.022 (Appendix A). This was unexpected, but similar results were recorded for *Salmonellae* carrying the *E. coli gyrA* gene alone (Figure 4, Appendix A). Thus, unlike *E. coli GyrB*, *E. coli* GyrA caused global supercoiling losses in *Salmonella*’s chromosome, and the biggest supercoil-losing positions were near genes with the highest transcription rates.

### 3.5. Transgenic Swaps of Salmonellae gyrB_St_ and gyrA_St_ in E. coli

We also tested *E. coli* strains carrying *Salmonella* gyrase subunits. However, the GyrB_St_ transgene swap was problematic. The frequency of colonies recovered on plates with GyrB_St_ linked to either Kn or Cm fell >100 fold relative to the efficiency of the reciprocal GyrB_Ec_ swap in *Salmonella*. In addition, when rare drug-resistant *E. coli* strains were analyzed, each strain had a transgene plus at least one copy of WT *gyrB_Ec_***_._** Our conclusion was that a preexisting duplication of GyrB_Ec_ was required to get a copy of GyrB*_S_*_T_. This confirms that the poor performance of this combination for in vitro supercoiling extends to the in vivo situation.

Nonetheless, results for GyrA_St_ swaps in *E. coli* were informative. Strains under selection for the *gyrA_St_* were recovered at a normal frequency, and there were no WT GyrA_Ec_ duplications. But, each transgenic *E. coli* strain required >2 days to become visible for picking on selective plates, and the cell doubling time on LB broth increased from 39 ± 1 min for the WT to 60 ± 3 min (Table 1). However, these transgenic strains (NH6380) exhibited constitutive RecA SOS induction, which made it impossible to measure supercoil densities in the chromosome. In RecA-SOS induced cells, background rates vary day to day and even from moment to moment because activated RecA induces cleavage of the lambda repressor that regulates our Resolvase plasmid. 

### 3.6. The GyrA CTD Controls Species Supercoiling

If the amino acid differences in GyrA_Ec_ cause transgenic sickness, is there a specific location of cause, or do multiple mutations located throughout the protein contribute to the phenotype? Assigned structure/function elements for GyrA are illustrated in Figure 5 with amino acid differences indicated by black lines above the map. The GyrA Winged Helix Domain (WHD) (purple) forms the DNA gate and includes the catalytic tyrosine (Y122). WHD is perfectly conserved between species and it interacts with the GyrB Topoisomerase-Primase domain (TOPRIM) Mg^++^ binding domain (green) and the C-terminal GyrA binding region that coordinates DNA cleavage and conformation changes that open and close the DNA gate [41]. The Tower domain (blue) forms the floor of the upper chamber, and the coiled coil domain forms the C-gate; these segments all have only have a few amino acid differences. Most divergence is localized in CTD, which includes six β-propeller elements (blue) and the red C-terminal acidic amino acid rich tail (Figure 5A).

The Berger lab showed that the acidic amino acid rich C-terminal tail of *E. coli* GyrA is responsible for different catalytic rates and enzymatic endpoints in comparative studies of the distantly related *M. tuberculosis* gyrase. Competition between the tail and DNA for binding to the CTD β-propellers (Figure 5A) [42,43] explains why the purified *E. coli* GyrA dimers do not bind DNA [33]. The *E. coli* GyrA tail has 14 substitutions and is three residues shorter than *Salmonella* GyrA tail (Figure 5A); these differences could explain why GyrA*_Ec_* is a more processive enzyme than GyrA*_S_*_T_. 

To evaluate the GyrA CTD contribution to chromosome supercoiling, chimeric *E coli* GyrA transgenes were introduced to *Salmonella*. In NH6392, the N-terminal 579 amino acids of *E. coli* GyrA were fused to 299 C-terminal residues of *Salmonella* GyrA at bp 1739. A second chimera in strain NH 6390 fused 705 N-terminal amino acids of *E. coli* GyrA to the last 173 amino acids of *Salmonella* GyrA at bp 2117. The first chimera includes the entire CTD of *Salmonella* GyrA, and the second chimera exchanges the last two-and-a-half pinwheel elements plus the acidic tail of *Salmonella* GyrA (Figure 5A, underlined purple line with the asterisk).

Supercoil assays at 10 loci revealed that WT supercoiling was restored (MIF =≥ 100) (Figure 5B) when *E. coli* GyrA was fused to the last 2 ½ β-pinwheels and tail elements of *Salmonella* GyrA (Figure 5B, purple numbers). A chimera with the *E coli* GyrA fused to the complete *Salmonella* GyrA CTD (NH6378) also restored WT levels throughout the genome (Appendix A and Figure 5B). The mean resolution efficiency increased 2.3-fold from 27 ± 15% to 62 ± 15%. This evidence demonstrates that the *Salmonella* GyrA CTD coordinates supercoil distribution throughout the *Salmonella* genome by a mechanism that is not obvious from quantitative in vitro supercoiling assays using a DNA substrate with a single highly processive gyrase binding site (Figure 2 and Figure 3).

### 3.7. Transgenic Salmonella GyrA Chimeras in E. coli

Two chimeras were made in *E. coli* to evaluate the cell stress phenotype of *Salmonella* GyrA in *E. coli.* A *gyrA_S_*_T_ transgenic strain NH6386 was modified twice. First, 579 N-terminal amino acids of GyrA_ST_ were fused to 296 C-terminal residues of GyrA_Ec_ in NH6451 (Figure 6A―Blue underline). This fusion is comparable to *Salmonella* chimera strain NH6390 and the results were similar. The slow division time of 60 min was restored to the WT value of 40 min (Table 1). A second fusion in *E. coli* NH6453 was an extreme test case: 840 N-terminal residues of *Salmonella* GyrA were spliced to only the last 35 amino acids of *E. coli* GyrA. The C-terminal 35 amino acids of *E coli* GyrA completely reversed cell sickness as this strain doubled in 37 ± 1 min (Table 1).

### 3.8. Chromosome Supercoil Density in E. coli

To evaluate supercoil distributions in the *E. coli* chromosome, supercoil sensors were inserted into four positions comparable to sites that were tested in *Salmonella* (Appendix A and Figure 6). Two sites are high transcription regions: the sensor at map position 83 min lies just upstream of the ATP operon and the sensor at 59 min is upstream of the *rrnG* operon. Two positions are not near known high transcription units in LB, including the *cynX* locus at 7 min and *ybfN* at 19 min. The average resolution efficiency for all four *E. coli* positions was 91 ± 5%; this is near the upper limit of our assay and it confirms a high diffusible supercoil density near σ _D_ = −0.042 (Appendix A).

The RecA-SOS phenotype disappeared in both the GyrA chimeras, along with a return to under 40 min doubling times. The γδ resolvase expression plasmid was well regulated in both, which allowed us to measure supercoiling relative to WT at four loci (Figure 6B, Appendix A). NH7001, NH7003, NH7004 and NH6999 have *Salmonella* GyrA fused to the complete *E. coli* CTD (Figure 6A). Their average supercoil density at 4 locations was 92 ± 3% (Figure 6B, blue numbers) which matched WT values (Figure 6B, black numbers). Strains NH6992, NH6994, NH6995, andNH6999 have only 32 amino acid changes back to match *E. coli* GyrA and they showed a resolution efficiency of 80 ± 4% (Figure 6B, red numbers). Therefore, the C-terminal 35 amino *E. coli* GyrA cures species-specific sickness and restores chromosome local supercoiling to near WT levels at 4 test positions (Figure 6B). GyrA CTDs in both species are crucial for recognizing species-specific signals at different DNA regions that establishes the WT supercoil chromosome domain structure.

## 4. Discussion

### 4.1. Evolution of Chromosomal Supercoil Control

*E coli* and *Salmonella* Typhimurium diverged from the “common ancestor” 120 million years ago in the mid Cretaceous age of dinosaurs [44]. Comparative study of these bacteria was historically focused largely on the different sets of phage and transposable sequences that give *E. coli* and *Salmonella* distinct biochemical advantages in their various niches. The core biochemical pathways of chromosome dynamics, DNA replication, transcription, and translation, have largely been assumed to be identical. To dissect the mechanism of GyrA control over supercoil levels, it has been necessary to focus on three factors. First, what conditions are known to alter gyrase supercoiling in living cells? Second, how are supercoil domains formed and organized in both species? Third, what role do strong gyrase sites play and how do they contribute to chromosome dynamics?

The first mechanism proposed to explain regulation of bacterial supercoil density was homeostatic control [45]. The theory was based on observations that transcription of *gyrA* and *gyrB* in *E. coli* increases when DNA supercoiling levels go down, while transcription of Topo I (the product of *topA*) increases when the supercoil level goes up. The model proposes that chromosomes attain a steady state by the mass action of these two enzymes that catalyze opposing reactions. Similar supercoil-sensitive transcription profiles for *gyrA, gyrB* and *topA* genes have been reported in *Haemophilus influenzae* [46], *Streptococcus pneumoniae* [47] and *Pseudomonas aeruginosa* [48]. However, the actual increase or decrease in these proteins were not measured at that time. When technology enabled investigators to carefully modulate expression of gyrase and Topo I in vivo [49], changing these protein levels had a small impact. A 10% increase or decrease in either gyrase or Topo I caused a 1.5% change in ambient plasmid supercoil density [50]. This value is at the detection limit of plasmid-based analysis. 

One factor that can change supercoil levels is connected to the energy source ATP. Gyrase binds both ATP and ADP, and the ratio of these two nucleotides influences supercoil endpoints in vitro; the higher the ATP/ADP ratio, the higher the supercoil density of a plasmid substrate. ATP/ADP ratios were found to vary in vivo when cells experience a number of stresses, including changing growth conditions. The ATP/ADP relationship explained plasmid supercoil density changes in cells shifted from low to high salt media [51], in cells shifted from aerobic to anaerobic growth conditions [52], and in *E. coli* cultures as they enter stationary phase [53]. However, the influence of ATP/ADP ratios is the same for *E. coli* and *Salmonella* gyrase in vitro.

### 4.2. Supercoil Domain Theory and Measurements

Another question that was initially addressed only in *E. coli* involved evaluation of the size and number of supercoil domains in the 4 Mb circular chromosome. *E. coli* was found to have 50 domains with a size of roughly 100 kb. Data for this calculation was collected from cells given X-ray treatment to introduce nicks in DNA, followed by a recovery period to seal nicks, followed by crosslinking to radioactive psoralen, which binds twofold better to supercoiled than to relaxed DNA [54,55,56]. However, when techniques were developed to measure supercoil domains in exponentially growing cells, the domain number in *E. coli* and *Salmonella* was higher by an order of magnitude. Different supercoil detection methods were developed for *Salmonella* [57,58] and *E. coli* [59], but results from both species showed 500 domains per genome equivalent with an average size of about 10 kb. Thus, 500 gyrase molecules per cell provides one gyrase for every domain, but there is not enough gyrase to ensure every domain gets one enzyme. An additional surprise noted in both experiments was that domain boundaries were not located at the same sites in every cell; rather, they were stochastic with respect to DNA sequence in different cells [57,59]. Boundaries also changed from moment to moment in the same cell [58]. The molecule(s) forming boundaries were a mystery. 

In 1987, a second mechanism that generates both (−) and (+) supercoils was proposed by Liu and Wang [60]. They argued that the movement of RNA polymerase along the DNA double helix would generate “twin domains” with (−) supercoils being generated behind a transcribed region and (+) supercoils being pushed ahead of RNAP. In 2004, Deng discovered that RNA polymerase does something unexpected by creating a supercoil barrier in DNA that blocks all supercoil diffusion across a transcribed region. A new supercoil boundary is established when transcription initiates, and the barrier disappears when gene expression is repressed [61]. The twin domain impact of RNA transcription was also confirmed in *Salmonella* in 2010. Booker showed that supercoil density is elevated behind the highly transcribed *rrnG* operon and supercoils are depleted in the domain downstream of the operon [8,12]. Thus, RNA polymerase not only supercoils DNA in vivo at rates comparable to gyrase, transcription also forms the most abundant supercoil domain boundaries in the bacterial cell [62]. 

Recent results demonstrating stochastic domain structure linked to transcription in *E. coli* came after high super-resolution microscopic techniques were developed for quantitatively measuring RNA synthetic rates at single DNA locations inside a single living cell [63]. Many genes studied using this technique showed a surprising transcription bursting pattern; an initial wave of transcription and translation was followed by strong inhibition, which could be followed at variable intervals by another burst. The phenomenon can’t be analyzed in cell populations because enzyme assays or microarray analyses only reveal population averages. 

Transcription bursting was subsequently modeled in vitro by firmly attaching covalently closed circular plasmids to a surface that blocks (+) and (−) supercoil diffusion from cancelling each other out [64]. Bursting in vivo and in vitro was explained by gyrase diffusion from one DNA site to the another. If gyrase binds to a domain with a transcription complex stalled by (+) supercoiling, a second burst occurs. When strong gyrase sites were added to plasmid bursting experiments along with gyrase, the bursting response was converted to a smooth expression pattern, which is the characteristic of *rrnG* operon and the 100 most highly transcribed operons in *E. coli* and *Salmonella* [12]. 

Since a transgenic *E. coli* GyrA subunit introduces different supercoil levels to different sites in the *Salmonella* chromosome, there must be signals in the genome at different locations that influences gyrase binding or activity. The simplest hypothesis is that chromosomal gyrase binding sites co-evolve in *E. coli* and *Salmonella* along with the enzyme. 

### 4.3. What Do Strong Gyrase Sites Look Like? 

Evidence that a single gyrase binding site can influence local DNA supercoiling came first from work on phage Mu [36]. Mu is a transposing replicon [65] that requires negative supercoiling to initiate transposition reaction [22]. When Mu grows on a TS GyrB mutant, it replicates poorly, making “pinpoint” plaques and generating very low phage yields. However, large plaque variants were isolated that produce phage titers on the TS GyrB strain that nearly match phage titers of the WT strain. Two phage mutants called *nuB1* and *nuB 103* [66] were found to elevate supercoil density in Mu while leaving the rest of the chromosome unchanged. The Mu SGS is a 150 bp region with three components that were dissected using genetics and biochemical analyses in the Pato lab [23,67,68,69,70]. The 150 bp Mu SGS includes a 40 bp left arm (Figure 7A brown), a 50 bp in the central core (Figure 7 Aqua), and a 60 bp the right arm (Purple). 

The gyrase DNA cleavage site is offset from center by 20 bp. Topological mapping shows that gyrase constrains 1 + supercoil which represents a 360° loop [71]. The right and left arms exhibit periodic hyper-sensitive sites at 10 bp intervals, reminiscent of the nucleosome DNase I pattern. With the center of DNA cleavage being defined as 0, the SGS left arm spans bp-65 to bp-26, but it lacks critical sequence or structure information since it can be deleted and adjacent sequences moved in to this position still support Mu replication. The critical G or gate segment (Figure 7A,B Aqua) spans DNA positions −25 to +25 and includes two 75° DNA bends that occur 1) at the 5’ to 3’ dinucleotide pair of +7G +8C of the top strand, and 2) at the 5’ to 3’ -7G -8C dinucleotide pair on the bottom strand. Both DNA bends are made by inserting an isoleucine into the minor groove; the same pattern is seen in high-resolution X-ray crystal structures of eukaryotic and prokaryotic type II topoisomerases [41]. The bends are also similar to DNA complexes with the chromosome associated protein IHF and HU, which place proline into the minor groove to create similar bending angles [72,73]. DNA bending aligns the L and R for interaction with the left and right pinwheel elements (Figure 6B).

The essential right arm of Mu’s SGS (purple) had DNAse I protection from bp +26 to bp +85 bp [70]. The first 40 nucleotides are essential for SGS function and include phased bending signals that optimize DNA looping over the pinwheels. This is similar to the nucleosome code where dinucleotide sequences promote optimal bending that creates phased nucleosomes in eukaryotes [76]. The transfer segment (green) can be any sequence; it stacks as a (+) chiral cross above the G segment (Figure 6) and is passed through the open DNA gate during catalysis (see below). The final requirement for the Mu SGS to promote Mu transposition is location. It only supports optimal Mu transposition when it is at the virus center; it promotes efficient left and right end synapsis prior to strand transfer [67]. 

### 4.4. A Species-Specific Rheostat Regulates Supercoiling at Strong Binding Sites

A “rheostat” model for processive supercoiling is illustrated for the Mu SGS in Figure 8. Supercoiling begins when DNA enters gyrase through the “ATP gate” (Step 1) and binds to the “floor” of the upper chamber (Figure 8B) with the aid of the strong 75° bends (Figure 7C). Competition between DNA and the acidic tails for binding pinwheel surfaces are reversible equilibria in the absence of ATP or a non-hydrolysable ATP analogue like AMP-PNP. ATP binding shifts the equilibrium to favor the DNA looping on the right pinwheel. Then in step 2 binding of ATP to both GyrB subunits closes the ATP gate and stimulates opening of the covalent gyrase-DNA gate, which passes the transfer region of DNA to the bottom chamber. Step 3 involves rigid body rotation of the GyrA and GyrB subunits that pass the transfer strand to the lower chamber where it can escape thorough the C Gate, and rejoining of the DNA break yields 2 supercoils. The energy for making supercoils comes from the binding energy of two ATPs, not from hydrolysis [77]. If the non-hydrolysable analog ADP-NP is added to a gyrase reaction, the substrate gains an increase of 2 (−) supercoils, but the DNA is trapped in a dead-end complex (red arrow). Hydrolysis of ATPs at step 4 re-opens the ATP gate, allowing the release DNA with an increase of two negative supercoils. 

Rheostat regulation occurs after step 4 when the pathway forks. One path (D) releases supercoiled DNA (Figure 3). The alternative path P proceeds immediately through another round of supercoiling. Rheostat control in *E. coli* GyrA involves 35 residues beyond the 6th pinwheel with 19 acidic residues, while the *Salmonella* rheostat has 38 amino acids beyond pinwheel 6 with 20 acidic units. *Salmonella’s* stronger tail favors DNA release while *E. coli’s* gyrase tail favors processive cycling. 

### 4.5. Where Are the Strong Gyrase Sites in E. coli and Salmonella chromosomes?

To our knowledge, there have been no focused searches to identify strong gyrase sites downstream of the top 50–100 highly transcribed operons in any organism. Families of 300 highly repetitive palindromes called Reps [78] or BIMEs [79,80] can bind gyrase with affinities at least 10-fold stronger than random DNA. But for many of these sequences, in vitro binding and cleavage analysis carried out on short DNA fragments gives complex patterns; 8 cleavage sites were identified in a single 344 bp Rep element [78]. But in vivo analysis of BIME gyrase cleavage patterns showed only one cleavage site per 2 repeats [81]. The importance of the arm sequence effect was unknown at the time of the early gyrase binding experiments, and BIMEs clearly have other functions, including the role of being loading sites on transcribed RNA for the transcription termination factor Rho [82]. Experiments using Next Generation Sequencing analysis of covalent GyrA-DNA complexes trapped by SDS lysis were recently reported for *E. coli* [83]. A similar study focused on *Salmonella* Typhimurium could answer many questions about species evolution and show precisely where gyrase binding is most critical for the class of highly transcribed operons. 

### 4.6. Gyrase Supercoiling Mechanisms in Distantly Related Species

How is supercoiling regulated in the more distantly related bacterial species? A supercoiling mechanism significantly different from the one shown in Figure 8 appears to be used in bacteria outside of the Gram-negative family. Unfortunately, chromosomal supercoil density measurements have not been made for most species, but it seems likely that DNA with linking numbers lower than *E. coli* could be the norm. The acidic tail of *E. coli*, which prevents GyrA dimers from stable binding to DNA [28], promotes fast supercoiling turnover and, when the tail is deleted, a very poor supercoiling reaction remains [34]. In the few Gram-positive organisms that have been studied to date, the GyrA tail is greatly minimized or completely missing. Bacillus subtilis has a reduced acidic residue C-terminus, and the enzyme supercoils with only a 2-fold penalty when it is deleted [35,84]. However supercoiling rates and endpoints haven’t been reported for most bacterial gyrases. The *Micrococcus luteus* GyrA binds double strand DNA quite well as a dimer and occupies specific locations on plasmid DNA, as measured by electron microscopy [85]. The acidic tail is absent in the GyrA homologues of *M. tuberculosis* [33] and *B. burgdorferi* [86]. For organisms lacking a tail, gyrase DNA binding may be controlled primarily by structural elements in the G region, which requires a strongly bent conformation [74]. 

### 4.7. E. coli GyrA in Salmonella

Why does the more powerful *E. coli* gyrase, and specifically the *E. coli* GyrA subunit, promote supercoil losses at locations in the *Salmonella* chromosome with the highest transcription rates, i.e., ATP and *rrnG* operons (Figure 4)? There are three non-exclusive mechanisms we can suggest to answer the mystery of GyrA toxicity. First, *Salmonella* gyrase has a stronger GyrA-tail, so a gyrase site downstream of the *Salmonella rrnG* and ATP operons may need to be strong sites to compensate for *Salmonellas’* weak enzyme. When the strong *E. coli* GyrA binds to a strong DNA site, the enzyme may be a road block to other enzymes that move on DNA, i.e., DNA polymerases, DNA helicases, DNA repair enzymes, and RNA polymerase. A second possibility involves a nonproductive ATP hydrolysis reaction that was discovered when *E. coli* gyrase was studied using relaxed ColE1 plasmid DNA, which includes a strong gyrase site to enhance site specific recombination [87]. In this case, *E. coli* gyrase continued to hydrolyze ATP after reaching the hyper-supercoil endpoint. This reaction is confusing, unproductive, and wasteful. If *E. coli* GyrA promoted a similar uncoupled ATP hydrolysis at 50 or 100 strong gyrase sites in a *Salmonella* chromosome at the rate of 5 ATPs /sec, the energy drain could lower ATP/ADP ratios, disrupt processes of DNA replication, proteins synthesis, and interfere general metabolism as a general energy sink. Third, a large percentage of chromosomal supercoiling is constrained by proteins that include the nucleoid-associated group of HU, H-NS, FIS, IHF, DPS, and STPA [88]. One or more of these proteins could directly or indirectly influence GyrA. An estimate of the ratio of unconstrained vs constrained supercoiling in plasmid DNA showed 60% constrained and 40% unconstrained supercoiling in *E. coli* [89]. This balance remained the same for WT, topA mutants, and gyrase mutants with altered the supercoil density within viable limits for *E. coli*. We assumed that this ratio also persists for *Salmonella* in the calculations shown in Appendix A. However, H-NS adopts several conformations including one that bridges two distant strands of DNA together [90,91]. The diffusion of supercoils catalyzed by gyrase near a strong patch of bridged DNA would restrict the ability of DNA to slither and branch. Supercoiling could be forced away from such locations. Recent biophysical studies show that the situation becomes even more complex. Every NAP listed above was analyzed by mass spec and found to have cellular modifications that include acetylation, succinylation, methylation, phosphorylation and deamidation at multiple positions [92]. Why these modified forms are made and where they are located in the chromosome is another significant housekeeping biochemical problem that should be investigated. 

## 5. Conclusions

The *E. coli* vs. *Salmonella* comparison turns out to provide a long list of results that challenge conventional theories of biochemical and genetic dogma. First, identical mutations in homologous “housekeeping” genes gyrA, gyrB, Topo I, bacterial condensin MukBEF, and the nucleoid associated protein H-NS, IHF, and HU, presented different phenotypes in *E. coli* and *Salmonella* [3]. These results forced us to deal with the fact that there is more than one way to use the same set of proteins to organize a bacterial chromosome and coordinate the complex connections between transcription, translation, and DNA replication [12]. In this chapter on DNA topology, the regulatory problem of coordinating supercoil equilibrium between the twin domains RNA transcription and gyrase turns out to be a local problem handled downstream of transcription termination. Two different C-terminal tails of GyrA are designed to deliver variable supercoil potential to the poorly and highly transcribed regions in both species, while rarely transcribed genes can burst and stall. Gyrase seems not designed to maintain a specific or ideal global average supercoil density, as many theories of regulation assume. Rather, the enzyme is designed for optimal support of RNA transcription that is important for growth in the most frequent niches of both species.

## Figures and Tables

**Figure 1 microorganisms-07-00081-f001:**
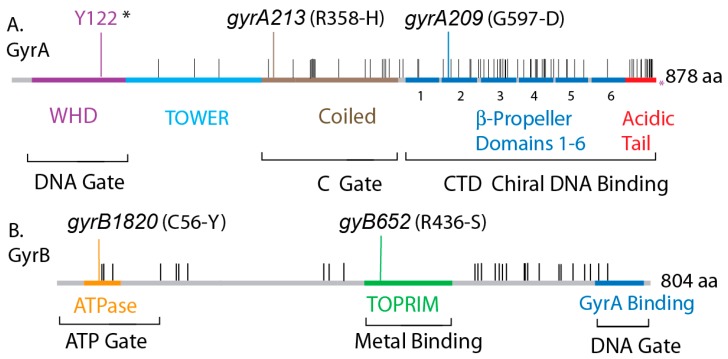
Conservation of protein sequences in *E. coli* and *Salmonella* gyrase and location of functional domains. (**A**) The GyrA protein is shown with the purple WHD domain, which contains the catalytic tyrosine (Y122*), the blue Tower domain, the brown coiled-coil domain that forms the C-gate, and the blue C-Terminal domain containing six pinwheel elements that fold into a spherical DNA binding surface. The red C-terminal acidic tail has many differences including the length of 38 residues in *Salmonella* compared to 35 in *E. coli*. (**B**) GyrB encodes the yellow ATPase domain that forms the ATP-gate, the green topoisomerase-primase (TOPRIM) domain that coordinates Mg++ binding, and a C-terminal GyrA interaction domain. Positions in GyrA and GyrB that encode different amino acids are shown as black hatches along the Salmonella map.

**Figure 2 microorganisms-07-00081-f002:**
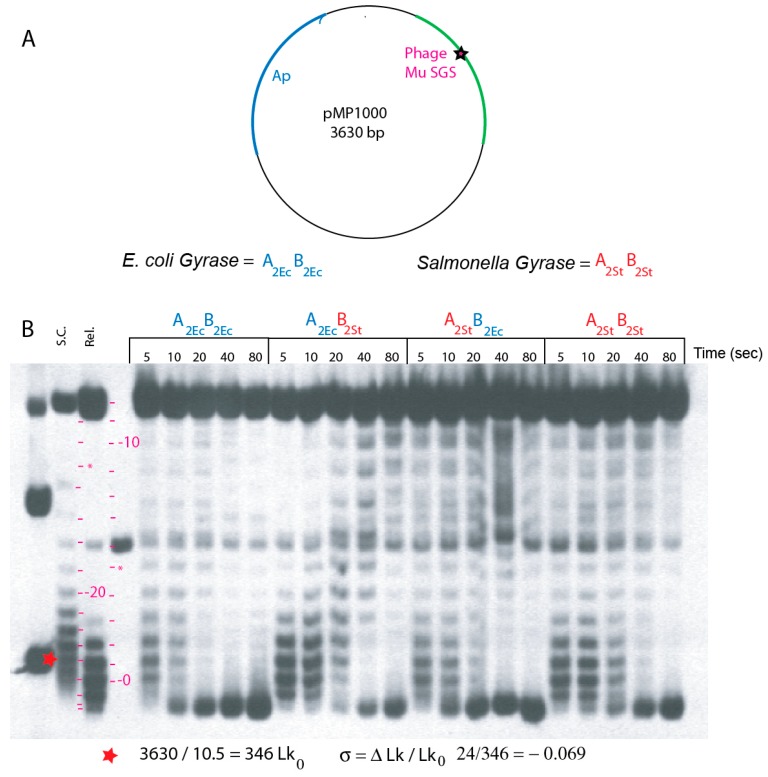
In vitro assays for supercoiling processivity in enzymes reconstituted from *E. coli* and *S. typhimurium* subunits. (**A**) The map of plasmid pMP1000 includes the nuB101 mutation of the phage Mu SGS that supports processive supercoiling reactions. (**B**) Southern blot patterns of plasmid DNA incubated with gyrases for different times (sec.) Supercoiled, relaxed, and linear pMP1000 plasmid was loaded in lanes **1**–**3** as markers. The positions of bands with linking numbers increasing from the relaxed substrate to −12 are shown by red dashes in lane **4**; topoisomers higher than −13 reverse and move faster down the gel at positions marked with red dashes between lanes **2** and **3**. DNA with higher supercoiling than the in vivo supercoiled species moves beyond the position of −24 *. Blue letters designate *E. coli* gyrase subunits; red letters designate *S. typhimurium* gyrase subunits.

**Figure 3 microorganisms-07-00081-f003:**
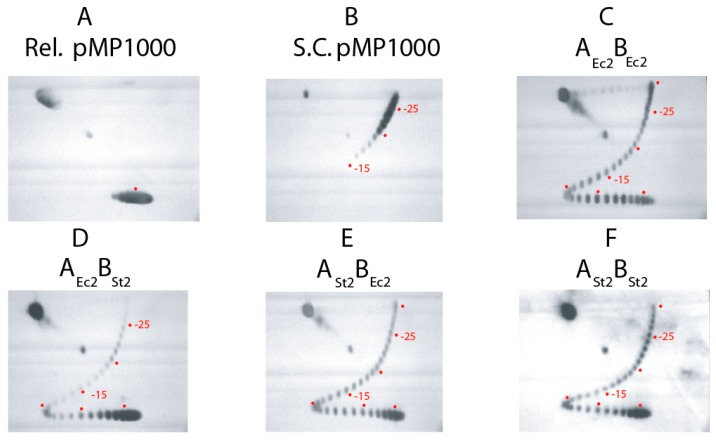
Two-D gel analysis of 5 sec assays for supercoiling of gyrase tetramers reconstituted with native partners and chimeric enzymes reconstituted with mixed *S. typhimurium* and *E. coli* subunits. The profile of Top1 relaxed DNA substrate and native supercoiled pMP1000 plasmid are shown in panels (**A** and **B**) respectively. Panels (**C**–**F**) show supercoil profiles of 5 s supercoiling reactions carried out with: *E. coli* gyrase, (**C**); chimera of *E. coli* GyrA and *Salmonella* GyrB, (**D**); chimera of *Salmonella* GyrA and *E. coli* GyrB, (**E**); *Salmonella* gyrase, (**F**). The positions of the center of the relaxed substrate is indicated when this band is visible, and positions of bands with −5, −10, −15, −20 and −25 supercoils are marked with red dots.

**Figure 4 microorganisms-07-00081-f004:**
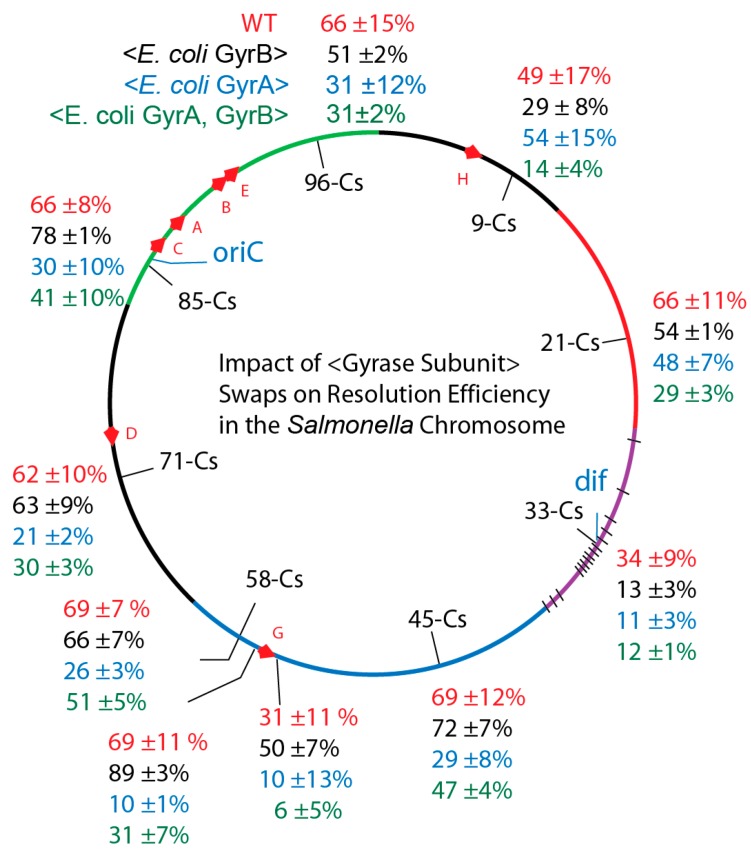
In vivo resolution assays of *Salmonella*e with precise chromosomal swaps of *E. coli* GyrA and GyrB subunits exploited supercoil reporters in six chromosome macrodomains. Macrodomain in *Salmonella* genome segments correspond to the six macrodomains of the *E. coli* chromosome: Ori-green; Right; Unstructured - black; Right, red; Ter, purple with black hatches showing *matS* sites; Left Domain, blue; Left unstructured domain, black. Resolution efficiencies at 10 positions are shown for: *Salmonella* WT, red; Strains with the *E. coli* GyrB transgene, black; Strains with *E. coli* GyrA, blue; and Salmonella with *E. coli* GyrA + GyrB, green.

**Figure 5 microorganisms-07-00081-f005:**
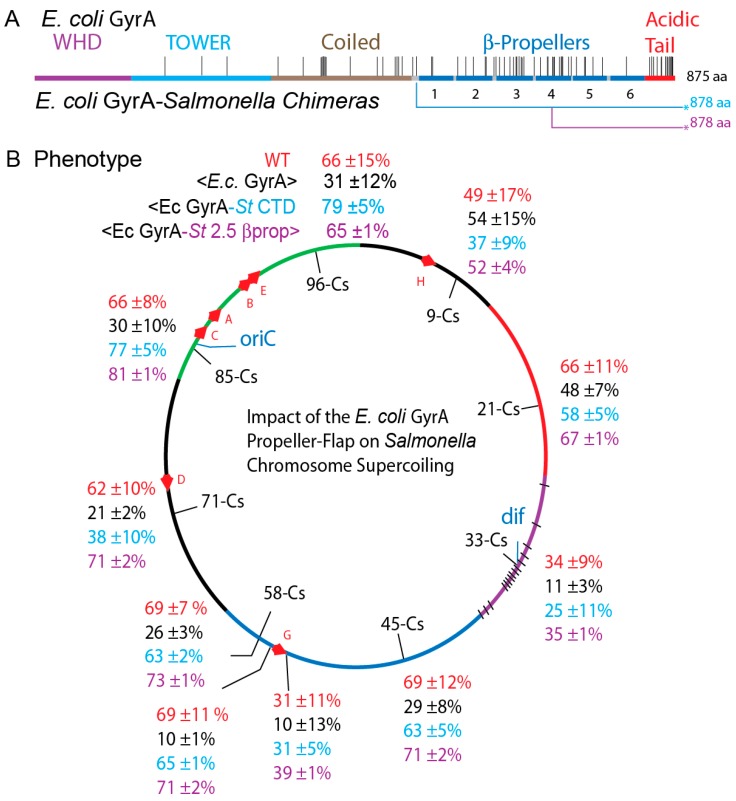
The C-terminal DNA binding domain of *E. coli* GyrA induces supercoiling losses in the *Salmonella* chromosome. (A) The genetic map of the different gyrase species tested is shown at the top (B) In vivo resolution results are compared for WT Salmonella gyrase―red numbers; strains with an *E. coli gyrA* replacing the *Salmonella* allele―black numbers; and two chimeras containing the N-terminal *E. coli* GyrA fused to *Salmonella* β-propeller elements at two points marked by blue and purple lines beneath the map.

**Figure 6 microorganisms-07-00081-f006:**
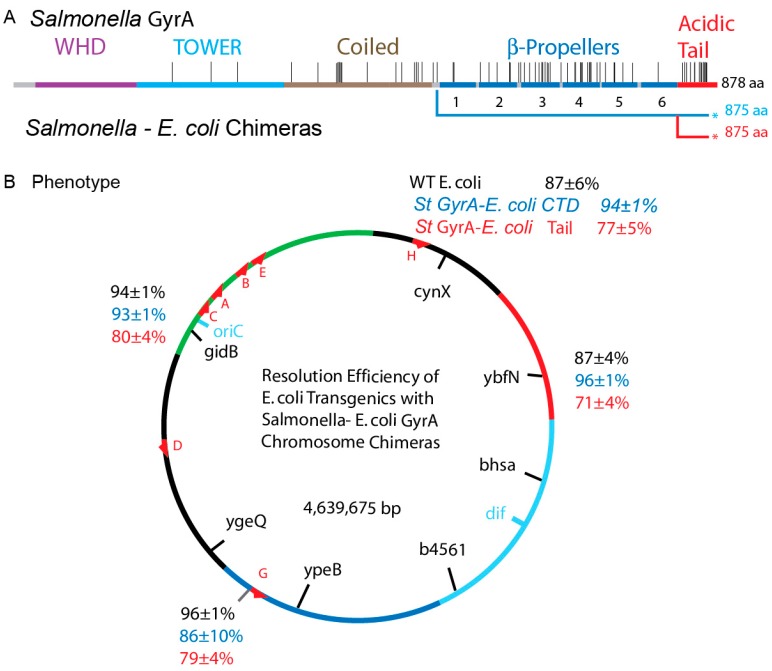
Resolution assays of WT *E. coli* and *E. coli* supercoiling with chimeric *Salmonella*–*E. coli* GyrA fusions. (A) A map of the genes that were tested. *E. coli* strains containing a WT GyrA, and two chimeric forms of *Salmonella-E. coli* GyrA were tested for supercoil-dependent resolution at four sites in the chromosome. (B) Resolvase deletion efficiencies for three gyrA genes are shown at 4 chromosomal locations. Strains with a WT GyrA and a Salmonella GyrA fused to the *E. coli* CTD at amino acid had very similar efficiencies at all four positions with a mean of 91 ± 5% and 92 ± 3% respectively. Strains carrying the *Salmonella* chimera fused to only the last 35 amino acids of the *E. coli* GyrA tail had a mean efficiency of 77 ± 4%, which shows a small (MIF) of 86.

**Figure 7 microorganisms-07-00081-f007:**
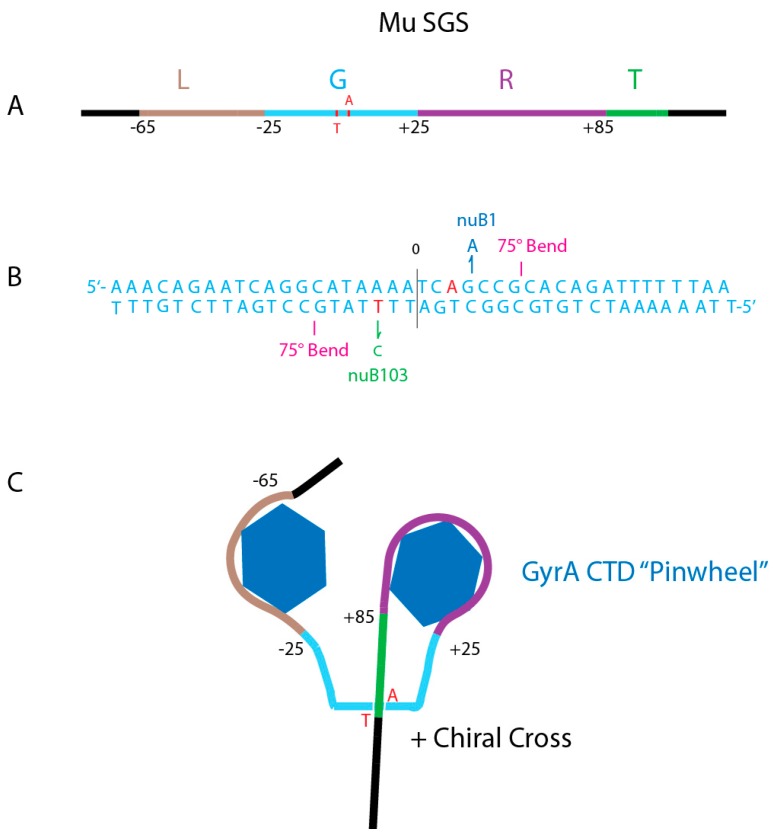
Anatomy of the Mu SGS. (**A**) Map of the 150 bp Mu SGS. Four regions include the left arm (Brown L), the Gate segment (Aqua G), the right arm (Purple R) and Transfer segment (Green T). Numbers show bp positions to the left (−) or to the right (+) of the center DNA cleavage. (**B**) The Gate DNA nucleotide sequence shows important positions for catalytic supercoiling. GyrA Tyr-122 makes a transient covalent bond with +3 A on the top strand and with −3 T on the bottom strand (red). Mutations that enhance supercoiling processivity include the +3 G-A transition (Blue nuB1) on top, and the *nuB103* −3 T to C transition (Green) on the bottom strand. Two 75° bends occur at positions +7G, +8C on top and at −7G, −8G on the bottom strand [74]. (**C**) Two CTD pinwheel elements of a GyrA dimer interact with L and R arms to make a (+1) loop that places the T segment in position to pass through an open gate during a sign inversion strand transfer [75].

**Figure 8 microorganisms-07-00081-f008:**
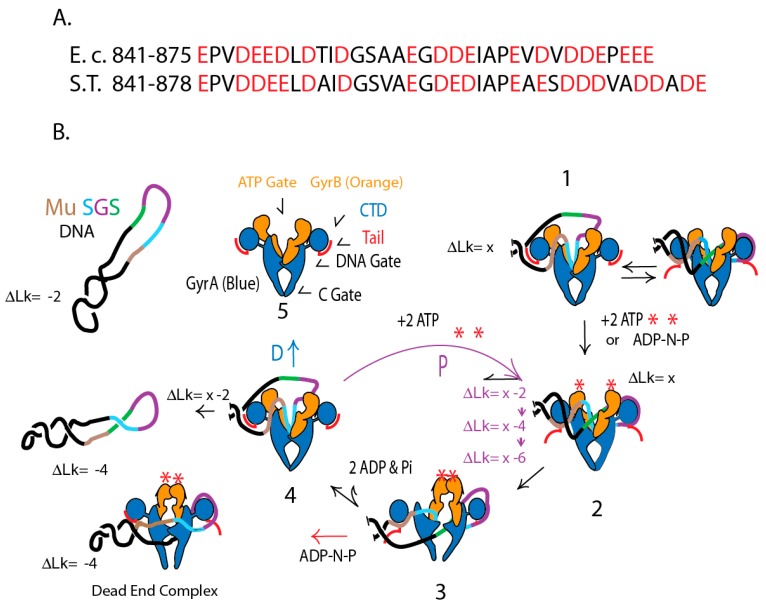
The GyrA red acidic tail controls of species-specific supercoiling. (**A**) Sequences of the GyrA C-terminal acidic amino acids (red) for *E. coli* and *Salmonella* Typhimurium. (**B**) Proposed mechanism to control supercoiling at different locations. Supercoiling is illustrated for a plasmid with a Mu SGS DNA binding site. The long arm (Brown) interacts with one gyrA pinwheel element, the gate segment (Aqua) folds into the upper chamber, and the right arm (Purple) interacts extensively with opposite pinwheel elements. The T segment (Green) passes through the open DNA gate during strand transfer. See text for step details.

**Table 1 microorganisms-07-00081-t001:** Doubling times of *S.* Typhimurium and *E. coli* strains with different gyrase subunits and chimeric gyrases containing GyrA fusions to whole CTD or 1/2 CDT domains of *Salmonella gyrA*.

Strain	GyrA	GyrB	Doubling Time (min) on LB 30 °C
*S.*Tm *NH6303*	*S.* Tm	*S.* Tm	40 ± 1
*S.* Tm *NH6304*	*S.* Tm	*E. coli*	43 ± 1
*S.T*m *NH6281*	*E. coli*	*S.*Tm	43 ± 1
*S.*Tm *NH6292*	*E. coli*	*E. coli*	44 ±1
*S.*Tm *NH6390*	*E. coli <*1/2 CTD *S.*Tm>	*S.*Tm	40 ± 2
*S.*Tm *NN6392*	*E. coli* <CTD *S.*Tm>	*S.* Tm	40 ± 1
*S.*Tm *NH6391*	*E. coli* <CTD *S.*Tm>	*E. coli*	39 ± 1
*E. coli NH1013*	*E. coli*	*E. coli*	39 ± 1
*E. coli NH6386*	*S. Tm*	*E. coli*	60 ± 3
*E. coli NH6451*	*S. Tm <*CTD *E. coli*>	*E. coli*	41 ± 1
*E. coli NH6453*	*S. Tm <*32 aa *E. coli* Tail>	*E. coli*	37 ± 1

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
