# Peer review of "Supercoil Levels in E. coli and Salmonella Chromosomes Are Regulated by the C-Terminal 35–38 Amino Acids of GyrA"

_microorganisms, 2019, doi:10.3390/microorganisms7030081_

Round 1

Reviewer 1 Report

In this manuscript the authors describe a comparative characterization of gyrase activity in vitro and in vivo in E. coli and Salmonella. These studies are motivated by the curious observation that despite being close relatives distinct supercoil densities are maintained in these species. The mechanisms underlying these differences are poorly understood.

The manuscript is well-written. The data reported are of good quality and generally presented in clear manner. This reviewer appreciates the efforts to obtain a cross-species biochemical understanding of enzymes, which is indeed receiving little attention, as well as the extensive reflective discussion section. The manuscript will be certainly of importance to the field.

Nevertheless, some small issues need to be addressed to further improve the clarity and readability of the manuscript:

Minor comments:

Line 107: it is not mentioned why a 2:1 ratio of GyrA:GyrB is used for reconstitution. Does it imply that in all in vitro experiments excess GyrA is present?

Legend Figure 1: ‘nucleosome-like’; maybe try ‘spherical’

Legend Figure 2 and main text: lane numbers are mentioned in the legend and main text, but lanes are not numbered in the figure

Line 211/Figure 3: the authors state that panels E and F demonstrate identical enzymatic activity (introduction of 5 supercoils/s) as seen in panel C. This does not seem correct. Already the activity seen in panel F appears somewhat lower than that in panel C, and the difference for panel E appears even larger

Figure 3: labelling of enzymes in text and figure is inconsistent (for instance in panel C: AEc2BEc2 instead of A2Ec-B2Ec in text)

Line 246: the authors define a Transgene Impact number for each locus at which supercoiling levels are measured. It is not clear to this reviewer why such term needs to be defined, as this number is nothing more but the ratio of the resolution efficiency of the transgene over that of the wt. The authors refer to tables S2, S3, S4 for an overview of measured efficiency efficiencies and TI numbers, but no TI numbers are shown in these tables. Instead MIF values are reported which seem to correspond to 1/TI. MIF seems to be reported once only in the main text (legend of figure 6).

Figure 4 and later figure reporting resolution efficiencies: do the differences in error in measured resolution efficiencies reflect biological differences?

Figure 1A/5A: annotation inconsistent (beta-propeller domains 1-6 vs beta-propellers)

Figure 8b: This figure is nice and useful, but currently the structure of this figure is not intuitive. Visual emphasis is currently on the left part, whereas step 1 is given at the top right. Also, not all parts of the figure are discussed in the text: e.g. the dead end complex is not mentioned.

Line 514/515: reference to path P seems to be missing in second part of this sentence

Lines 528-531: it may be nice to include a discussion of the results reported in the very recent study of Sutormin et al., Nucl Acids Res, 2018/19 (Single-nucleotide-resolution mapping of DNA gyrase cleavage sites across the Escherichia coli genome)

Typographical issues:

-       Line 52: decatnates à decatenates

-       Line 58: lower that à lower than that of

-       Line 97: Quiagen à Qiagen

-       Line 112: routinely – term not needed

-       Line 275: invitro à in vitro

-       Line 280: downstream the rate à downstream rate

-       Line 293: remove ‘that only cells’

-       Line 299: strais à strains

-       Line 331: CDTà CTD

-       Line 414: supercoils à supercoil

-       Line 526: were à was

-       Line 539: Gram-Positive à Gram-positive

-       Line 544: M. Tuberculosis à M. tuberculosis

-       Line 547: try ‘question why’ instead of ‘reason’

-       Line 575: appearà apppears

Author Response

I want to thank this reviewer for their excellent and fast critique of our paper.  This was a very nice introduction to this new journal.

Minor comments:

Line 107: it is not mentioned why a 2:1 ratio of GyrA:GyrB is used for reconstitution. Does it imply that in all in vitro experiments excess GyrA is present? Yes.

Response: Historically, in vitro experiments with gyrase were routinely reconstituted a 2-fold excess of GyrA .  The reason is that it reflects the ratio of GyrA to GyrB subunits in living E. coli cells where you find twice as much GyrA dimers as GyrB to form tetramers (27).  GyrB is also less stable that GyrA, so the problem is always running out of GyrB.  No matter how you reconstituted gyrase with equal molar subunits or with an excess of GyrB, you still only get the A2-B2 tetramer binding to DNA. But  when GyrB runs out and you need to do another 300 L purification to get another lot of GyrB and you  pile up even more GyrA.  What might this means in vivo?  When cells need more gyrase, they only have to express GyrB to double the cellular enzyme level.  GyrB is also transcribed at higher levels that GyrA when the DNA  (20).  The following section was added to explain how time course experiments are set up.

Line 109

 Preliminary experiments were always carried out before a time course series in which serial dilutions of the each GyrA-GyrB  pair to be tested were serially diluted, pre-incubated with relaxed DNA for 30 min at 30°, and then incubated after ATP addition for 90 seconds.  The highest dilution that supercoiled all of the relaxed substrate was then used for time course studies on the same day.  The reason that GyrB is the limiting subunit is that it is present in cells at half the concentration of GyrA [27]. TS GyrB preparations are especially troublesome; they are easily denatured by going out of and back into a -20° freezer and must often be purified fresh for critical experiments.

Legend Figure 1: ‘nucleosome-like’; maybe try ‘spherical’

Response: excellent suggestion, we changed to spherical (now on line 146)

Legend Figure 2 and main text: lane numbers are mentioned in the legend and main text, but lanes are not numbered in the figure

I struggled with this figure and tried to include lane numbers below, but this made a complex figure even worse.  Since both reviewers recognized this problem I decided to explain the results as four easily recognized time course blocks.  I hope this solves their problem because as someone who suffers from dyslexia, I get constantly confused by looking for lane number. This figure is really a prelude to the next figure which is much less cluttered and easier to understand.  

Line 211/Figure 3: the authors state that panels E and F demonstrate identical enzymatic activity (introduction of 5 supercoils/s) as seen in panel C. This does not seem correct. Already the activity seen in panel F appears somewhat lower than that in panel C, and the difference for panel E appears even larger  To be clear we state the power order, now on line 222-223 “Overall, the power relationship was AEc2-BEc2> AST2-BST2  > ASt2-BEc2 > AEc2-BSt2.”

Figure 3: labelling of enzymes in text and figure is inconsistent (for instance in panel C: AEc2BEc2 instead of A2Ec-B2Ec in text)

The  inconsistencies were corrected in lines 219-221.

Line 246: the authors define a Transgene Impact number for each locus at which supercoiling levels are measured. It is not clear to this reviewer why such term needs to be defined, as this number is nothing more but the ratio of the resolution efficiency of the transgene over that of the wt. The authors refer to tables S2, S3, S4 for an overview of measured efficiency efficiencies and TI numbers, but no TI numbers are shown in these tables. Instead MIF values are reported which seem to correspond to 1/TI. MIF seems to be reported once only in the main text (legend of figure 6).  

All impact factors are now referred to as MIF as they are in the tables.

Figure 4 and later figure reporting resolution efficiencies: do the differences in error in measured resolution efficiencies reflect biological differences?

It is probable, because the wide variations at certain loci are repeatable.  These ensemble measurements have cells in all stages of growth and the issue of variations at certain locations can not be addressed at the present time. 

Figure 1A/5A: annotation inconsistent (beta-propeller domains 1-6 vs beta-propellers)

All are now called beta-propeller elements

Figure 8b: This figure is nice and useful, but currently the structure of this figure is not intuitive. Visual emphasis is currently on the left part, whereas step 1 is given at the top right. Also, not all parts of the figure are discussed in the text: e.g. the dead end complex is not mentioned.

 The missing part is now explained to show that binding energy alone fuels the supercoiling process and hydrolysis is only needed to release DNA.

Line 737

Step 3 involves rigid body rotation of the GyrA and GyrB subunits that pass the transfer strand to the lower chamber where it  can  escape thorough the C Gate, and rejoining of the DNA break to yield 2 supercoils. The energy for making supercoils comes from the binding energy of two APT, not from hydrolysis [76]. If the non-hydrolysable analog ADP-NP is added to a gyrase reaction, the substrate gains an increase of 2 (-) supercoils, but the DNA is trapped in a dead-end complex (red arrow). Hydrolysis of  ATPs at step 4 re-opens the ATP gate allowing the release DNA with an increase of two negative supercoils.

Line 514/515: reference to path P seems to be missing in second part of this sentence

It was added to the discussion along with an explanation of the non-hydrolyzable reaction with APPnP in lines

Lines 528-531: it may be nice to include a discussion of the results reported in the very recent study of Sutormin et al., Nucl Acids Res, 2018/19 (Single-nucleotide-resolution mapping of DNA gyrase cleavage sites across the Escherichia coli genome) 

785 Experiments using Next Generation Sequencing analysis of covalent GyrA-DNA complexes trapped by SDS lysis were recently reported for E. coli[82]. A similar study focused on SalmonellaTyphimurium could answer many questions about species evolution and show precisely where gyrase binding is most critical for the class of highly transcribed operons.

Thank you thank you.  We added this information to the 

Typographical issues:

 All were corrected as suggested

-       Line 52: decatnates à decatenates 

-       Line 58: lower that à lower than that of

-       Line 97: Quiagen à Qiagen - now line 98

-       Line 112: routinely – term not needed deleted

-       Line 275: invitro à in vitro

-       Line 280: downstream the rate à downstream rate now line 316

-       Line 293: remove ‘that only cells’ deleted

-       Line 299: strais à strains now 339

-       Line 331: CDTà CTD now 377

-       Line 414: supercoils à supercoil 

-       Line 526: were à was

-       Line 539: Gram-Positive à Gram-positive

-       Line 544: M. Tuberculosis à M. tuberculosis now line 620

-       Line 547: try ‘question why’ instead of ‘reason’ Excellent suggestion now on line 623

-       Line 575: appearà appears

Reviewer 2 Report

Review of microorganisms-459561 Rovinskiy et al.

This manuscript describes a comprehensive analysis of the perplexing difference in steady-state negative supercoiling in E. coli (higher supercoil density) and Salmonella (15% lower).  The authors approach the problem by thoughtfully dissecting the tetrameric gyrase enzymes from the two organisms, creating mixed gyrases in vitro and in vivo, and then chimeric gyrases to assess which aspects of gyrase structure may tune global supercoiling. They find in vitro that the fully E. coli enzyme is more active than the Salmonella enzyme, which is comparable to the mixed enzyme with E. coli B subunits, all of which are more active than the mixed enzyme with Salmonella B subunits.  In vivo, Salmonella does not grow well with E. coli gyrase and E. coli does not tolerate Salmonella gyrase, all despite relative similarity of the enzymes. The authors map supercoil densities across the chromosome for these and other mutants. Interestingly, they show that it appears to be the C-terminal domain (i.e. final 35-38 amino acids) that determines the supercoiling level. Specifically, a subtle difference in the number of acidic amino acids is suggested as an explanation and a model in which the ability of the acidic tail of the A subunit of gyrase to compete with DNA binding determines processivity, such that the less (by one charge) acidic tail of E. coli gyrase A subunit causes less competition and provokes DNA binding of longer duration.

This is an impressive study, addressing an interesting problem, and suggesting a surprising explanation. Especially nice is the historical tone and didactic approach inherent in the work.

Points for consideration by the authors

1. It would be helpful to clarify within the manuscript when measurements reflect unrestrained vs. total (restrained plus unrestrained) supercoiling. In vivo measurements presumably reflect the former. Is there concern that for strains suffering from gyrase dysfunction, expression of many genes may be altered, including those affecting levels of DNA-wrapping proteins that alter restrained supercoiling. Perhaps the supercoil sensor device used by the authors is immune from such effects, but some comment is encouraged.

2. The authors focus their models and explanations on the notion that the various gyrases have different affinities for specific DNA sequences (thus accounting for hot spot gyrase binding sequences that might differ between strains), and altered gyrase subunit composition may change such DNA affinities (see for example lines 448-451). This reasoning is fine, as far as it goes, but it does not address the possibility that gyrase affinity is also influenced by binding cooperativity between gyrase and adjacent sequence-specific proteins bound to DNA. As the authors well know, cooperativity between DNA-bound protein is a fundamental concept in explaining apparent DNA affinity for many DNA binding proteins that can favorably interact as well with other DNA-bound proteins. It should be mentioned that many of the effects described here might be influenced by species-specific differences in cooperative interactions between gyrase and local DNA binding proteins unique to each species. While the authors propose a model involving competition between DNA and the anionic C-terminal domain, the actual difference in charge is subtle. The authors should discuss the possibility that it is differences in cooperative binding interactions between different versions of gyrase and other DNA binding proteins that may suffer when inter-species mixed gyrases are created. Thus the question “What do strong gyrase sites look like?” (page 13) might include the possibility that they are sites allowing favorable cooperative contact between gyrase and other sequence-specific DNA binding proteins. Species differences might involve cooperativity depending on the C-terminal tail region. In that regard, it should be noted that the in vitro gyrase activities of the CTD chimeras were not tested in vitro in this work, making it difficult to parse the specific activity of these chimeras vs. the potential effects on potential cooperative interactions.

Smaller points

1. Page 3 methods, line 107. Why is gyrase reconstituted with GyrA:GyrB in 2:1 ratio when the enzyme is a 2:2 tetramer? Readers will wonder.

2. Figure 2: please add lane numbers below the image.

2. The authors do not mention it here, but how does their chromosome supercoil sensor (supplemental figure 1) depend on lac induction, e.g. by IPTG? Is it the repressed form of the lac promoter, or the active form that most alters slithering? Thus, is the sensor measuring only unrestrained supercoiling in the region of the resolvase, or also barriers to slithering?

Author Response

We appreciate this very prompt and thoughtful review.  Our responses are shown below

1.    It would be helpful to clarify within the manuscript when measurements reflect unrestrained vs. total (restrained plus unrestrained) supercoiling. In vivo measurements presumably reflect the former. Is there concern that for strains suffering from gyrase dysfunction, expression of many genes may be altered, including those affecting levels of DNA-wrapping proteins that alter restrained supercoiling. Perhaps the supercoil sensor device used by the authors is immune from such effects, but some comment is encouraged.

Response: We now include a discussion of constrained and unconstrained supercoils and how it was estimated and used in the tables 2 and 3. 

Line 819

Third, a large percentage of chromosomal supercoiling is constrained by proteins that include the nucleoid-associated group of  HU, H-NS, FIS, IHF, DPS, and STPA [87].  One or more of these proteins could directly or indirectly influence GyrA.  An estimate of the ratio of unconstrained vs constrained supercoiling in plasmid DNA showed 60% constrained and 40% unconstrained supercoiling in E. coli[88]. This balance remained the same for WT, topA mutants, and gyrase mutants with altered the supercoil density within viable limits for E. coli.  We assumed that this ratio also persists for Salmonellain the calculations shown in Table S2 and S3. 

2. The authors focus their models and explanations on the notion that the various gyrases have different affinities for specific DNA sequences (thus accounting for hot spot gyrase binding sequences that might differ between strains), and altered gyrase subunit composition may change such DNA affinities (see for example lines 448-451). This reasoning is fine, as far as it goes, but it does not address the possibility that gyrase affinity is also influenced by binding cooperativity between gyrase and adjacent sequence-specific proteins bound to DNA. As the authors well know, cooperativity between DNA-bound protein is a fundamental concept in explaining apparent DNA affinity for many DNA binding proteins that can favorably interact as well with other DNA-bound proteins. It should be mentioned that many of the effects described here might be influenced by species-specific differences in cooperative interactions between gyrase and local DNA binding proteins unique to each species. While the authors propose a model involving competition between DNA and the anionic C-terminal domain, the actual difference in charge is subtle. The authors should discuss the possibility that it is differences in cooperative binding interactions between different versions of gyrase and other DNA binding proteins that may suffer when inter-species mixed gyrases are created. Thus the question “What do strong gyrase sites look like?” (page 13) might include the possibility that they are sites allowing favorable cooperative contact between gyrase and other sequence-specific DNA binding proteins. Species differences might involve cooperativity depending on the C-terminal tail region. In that regard, it should be noted that the in vitro gyrase activities of the CTD chimeras were not tested in vitro in this work, making it difficult to parse the specific activity of these chimeras vs. the potential effects on potential cooperative interactions.

 The following point was added about the importance of NAPs

819 Third, a large percentage of chromosomal supercoiling is constrained by proteins that include the nucleoid-associated group of  HU, H-NS, FIS, IHF, DPS, and STPA [87]……However, H-NS  adopts several conformation including one that bridges two distant strands of DNA together [8990].  The diffusion of supercoils catalyzed by gyrase near a strong patch of bridged DNA would restrict the ability of  DNA to slither and branch. Supercoiling could be forced away from such locations.  Recent biophysical studies show that the situation becomes even more complex.  Every NAP listed above was analyzed by mass spec and found to have cellular modifications that include acetylation, succinylation, methylation, phosphorylation, and deamidationat multiple positions [91]. Why these modified forms are made and where they are located in the chromosome is another significant housekeeping biochemical problem that should  be investigated. 

Smaller points

1. Page 3 methods, line 107. Why is gyrase reconstituted with GyrA:GyrB in 2:1 ratio when the enzyme is a 2:2 tetramer? Readers will wonder.

 Response: Historically, in vitro experiments with gyrase were routinely reconstituted a 2-fold excess of GyrA .  The reason is that it reflects the ratio of GyrA to GyrB subunits in living E. coli cells where you find twice as much GyrA dimers as GyrB to form tetramers (27).  GyrB is also less stable that GyrA, so the problem is always running out of GyrB.  No matter how you reconstituted gyrase with equal molar subunits or with an excess of GyrB, you still only get the A2-B2 tetramer binding to DNA. But  when GyrB runs out and you needed to do another 300 L purification to get another lot of GyrB and you  pile up even more GyrA.  What might this means in vivo?  When cells need more gyrase, they only have to express GyrB to double the cellular enzyme level.  GyrB is also transcribed at higher levels that GyrA when the DNA  (20).  

The following section was added to explain how time course experiments are set up.

Line 109

 Preliminary experiments were always carried out before a time course series in which serial dilutions of the each GyrA-GyrB  pair to be tested were serially diluted, pre-incubated with relaxed DNA for 30 min at 30°, and then incubated after ATP addition for 90 seconds.  The highest dilution that supercoiled all of the relaxed substrate was then used for time course studies on the same day.  The reason that GyrB is the limiting subunit is that it is present in cells at half the concentration of GyrA [27]. TS GyrB preparations are especially troublesome; they are easily denatured by going out of and back into a -20° freezer and must often be purified fresh for critical experiments.

2.    Figure 2: please add lane numbers below the image.

Response: This section was rewritten and I focused on the four time clusters rather than making lane designations.  I tried the idea of putting lanes along the bottom, but this created a very dense and confusing (for me) job of writing what to look for. 

3.    The authors do not mention it here, but how does their chromosome supercoil sensor (supplemental figure 1) depend on lac induction, e.g. by IPTG? Is it the repressed form of the lac promoter, or the active form that most alters slithering? Thus, is the sensor measuring only unrestrained supercoiling in the region of the resolvase, or also barriers to slithering?

Response When we use the sensor to measure supercoiling, the complete lac operon in the element must be repressed. However, it can be used to measure coupled transcription/translation rates around the chromosome at points of known supercoil density and to evaluate the general effects of all the NAPs on coupled transcription/translation in vivo.